# The Devil is in the Edges: Monocular Depth Estimation with Edge-aware Consistency Fusion

## Abstract

This paper presents a novel monocular depth estimation method, named ECFNet, for estimating high-quality monocular depth with clear edges and valid overall structure from a single RGB image. We make a thorough inquiry about the key factor that affects the edge depth estimation of the MDE networks, and come to a ratiocination that the edge information itself plays a critical role in predicting depth details. Driven by this analysis, we propose to explicitly employ the image edges as input for ECFNet and fuse the initial depths from different sources to produce the final depth. Specifically, ECFNet first uses a hybrid edge detection strategy to get the edge map and edge-highlighted image from the input image, and then leverages a pre-trained MDE network to infer the initial depths of the aforementioned three images. After that, ECFNet utilizes a layered fusion module (LFM) to fuse the initial depth, which will be further updated by a depth consistency module (DCM) to form the final estimation. Extensive experimental results on public datasets and ablation studies indicate that our method achieves state-of-the-art performance.

## 1 Introduction

Monocular depth estimation (MDE) is a classical and fundamental computer vision task with a wide range of applications in autonomous driving, the metaverse, and robotics. Without the depth data from depth sensors or geometric constraints from the posed multi-view images, it is tricky to recover the 3D structure of the observed scene from a single image. Recently, learning-based methods Bhat et al. (2021); Laina et al. (2016); Yin et al. (2021b); Ranftl et al. (2022); Wu et al. (2022) demonstrate great potential in this task by using networks, (*e.g.*, CNN Ranftl et al. (2022) or transformers Ranftl et al. (2021)) to map the RGB colors to depth values. Although various network architectures Johnston & Carneiro (2020); Poggi et al. (2020); Fu et al. (2018), loss functions Yin et al. (2021a); Xian et al. (2020), and training strategies Chen et al. (2020a); Godard et al. (2017); Wong & Soatto (2019) have been proposed to improve the estimated depth, existing methods still suffer from missing details and over-smooth estimations in the final depth. To handle this problem, several works exploit the edge information Yang et al. (2018) and image segmentation Zhu et al. (2020) to enhance the depth details. Although these methods have made progress in predicting depth maps with details, the estimated depth in edges is still struggling, leading to over-smooth final predictions. Estimating high-quality depth maps with accurate edges and details remains a significant obstacle for MDE networks.

This paper aims to reduce the ambiguity in the important edges and detailed regions, as well as acquire finer overall depth predictions. We ask: *what has the greatest impact on edge depth*? We perform various experiments and arrive at a conjecture: *the edge itself hides the most critical information*. The following analysis demonstrates our proposal from several perspectives.

(**i**) By inspecting the predicted depths of the existing MDE networks Ranftl et al. (2021; 2022); Yin et al. (2021b), we find that they can obtain competent results in obvious and large edge regions, (*e.g.*, edges of large foreground objects), but their predictions in low-contrast edges, long distance, and small edges are very fuzzy and inaccurate as shown in Fig. 1. We analyze and attribute this phenomenon to that networks do not capture all the edge information as the heavily-used convolution layers, downsampling and upsampling operations would smooth out small natural edge areas. To validate this ratiocination, we test the same pretrained MDE network Ranftl et al. (2021) with different inputs: the original images, the corresponding edge maps, and the images with highlighted edges (by deleting edge pixels). As shown in Fig. 1, the edge maps and edge-highlighted images obtain clearer

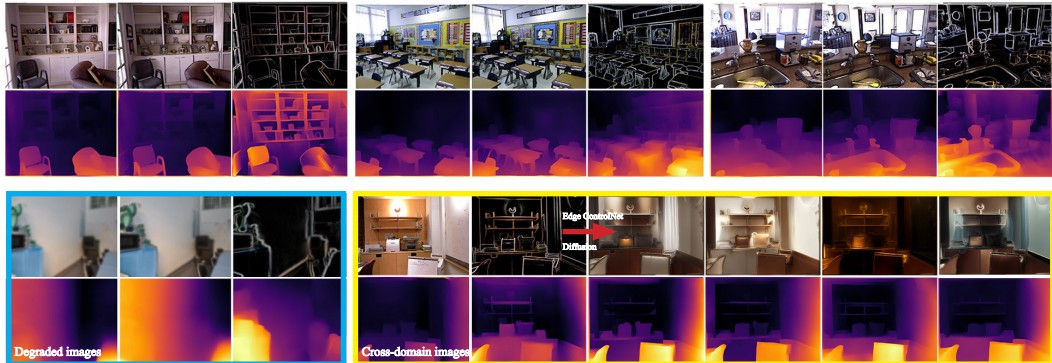

Figure 1: Depth visualizations on NYU-v2 Silberman et al. (2012). For each triplet in the firsr row, we showcase the RGB image, the edge-highlighted map, the edge map, and their corresponding depth maps predicted by pre-trained DPT Ranftl et al. (2021). The edge maps are obtained using the hybrid edge detection strategy. In the second row, we show more edge observations in degraded images or cross-domain applications.

edges (please ignore the structural distortion in the depth maps). The aforementioned results indicate that the edge itself plays a critical role in producing fine details and edges.

(**ii**) Another phenomenon that supports our conjecture is the performance drop of the degraded and cross-domain images. Some existing works Hu et al. (2019); Dijk & Croon (2019) believe that MDE networks rely on geometric cues, occlusion boundaries, and textures to produce depth details. However, they cannot explain the significant performance gap between the normal and degraded images, as these factors would hardly be affected by the slight noise. Therefore, we attribute the performance drop to the disturbance of the edge information, which is more than sensitive to noise or blur. As illustrated in Fig. 1(blur box), we compare the predicted depths on the degraded images, the corresponding edge maps, and the edge-highlighted images, the latter two inputs turn out to be robust to the image degradation, which suggests the key to producing clearer edges is to preserve and utilize the edge information. Furthermore, as shown in the yellow box, we utilize ControlNet's Zhang & Agrawala (2023) edge control conditions and stable diffusion models Rombach et al. (2022) to generate a series of images with diverse styles from the edge map. Despite the variations in texture and color information among these images, they consistently produce nearly identical depth maps. We attribute this phenomenon to the consistency in their edge structure information. Please see more details in the Appendix A.1.

Driven by this analysis, we propose to dig into the edge information and use it to produce high-quality depth maps with accurate edges and details. Specifically, we present an Edge-aware Consistency Fusion network (ECFNet) which mainly consists of two parts: Layered Fusion Module (LFM) and Depth Consistency Module (DCM). LFM plays a role in acquiring clear edges by fusing the initial depth maps from the edge maps, the edge-highlighted images, and the original depth maps. The key idea of LFM derives from the fact that the edge itself helps predict sharper depth edges (see Fig. 1). As the quality of the detected edges would affect the fused depth, we develop a hybrid edge detection strategy to get high-quality edges, which incorporates the classical and learning-based edge detection methods. However, as the edge maps contain no texture and shadow cues, the corresponding predicted depth maps suffer from the wrong spatial structure and inaccurate scale. What's more, even the same fixed MDE network can not guarantee the predictions from different inputs, (*i.e.*, the original image, the corresponding edge map, and the edge-highlighted image) have the same depth range and distribution. The aforementioned two problems lead to a thorny phenomenon: the fused depth maps from LFM have clear edges but inaccurate overall structure. To handle this problem, we propose DCM to learn the local depth residual between the fused depth and the initial depth. The learned residual will be further used to update the fused depth. The goal of DCM is to maintain the high-frequency details and reduce the structural errors in the fused depth, which is critical to get the high-quality final depth.

We conduct various experiments to compare ECFNet with other methods and to validate the effectiveness of different modules. Extensive experimental results indicate that ECFNet achieves

state-of-the-art performance compared to related methods, especially in producing accurate edge depth. We also illustrate that ECFNet achieves significantly superior performance to the existing *SOTA* methods when faced with degraded images.

Our contributions can be summarized as follows:

- We explore and validate that edge information plays a critical role in producing high-quality depth maps with clear edges and details, providing new insights for cross-domain downstream tasks utilizing clear edge depth maps (see details in Appendix A.2).
- We present a novel edge-aware consistency fusion network (ECFNet) to predict accurate monocular depth, which consists of a layered fusion module and a depth consistency module.
- Experimental results on multiple datasets indicate that our method achieves state-of-the-art performance.

## 2 RELATED WORKS

**Single image depth estimation**    Monocular depth estimation aims to predict depth from a single RGB image. The existing works usually learn a non-linear mapping from pixels to depth values through deep neural networks Walia et al. (2022); Eigen et al. (2014); Bhat et al. (2021); Laina et al. (2016); Yin et al. (2021b); Ranftl et al. (2022); Wu et al. (2022). Remarkable progress in MDE has been witnessed in recent years, such as the large-scale depth datasets Chen et al. (2020b); Kim et al. (2018); Niklaus et al. (2019); Li & Snavely (2018), the novel neural architecture designs Fang et al. (2020); Johnston & Carneiro (2020); Poggi et al. (2020), the various loss functions Yin et al. (2021a); Xian et al. (2020); Yin et al. (2021b); Lee & Kim (2020), and the efficient training strategies Ranftl et al. (2021; 2022); Chen et al. (2020a); Godard et al. (2017); Wong & Soatto (2019); Li et al. (2021).

Nevertheless, existing MDE models Ranftl et al. (2021); Yin et al. (2021b); Ranftl et al. (2022); Xian et al. (2020) often fail to predict depth maps with abundant high-frequency details, especially when encountered with degraded images. On the contrary, our approach enhances these details by incorporating the depth map generated from the edge map, and thus is able to predict clear and continuous edges in the depth map even in the degraded images.

**Edges in depth estimation**    Some recent works notice that the edge information would be helpful in training depth estimation networks Zhu et al. (2020); Talker et al. (2022); Wang et al. (2020); Yang et al. (2018); Ramamonjisoa et al. (2020); Qiu et al. (2020). Zhu *et al.* Zhu et al. (2020) propose to regularize the depth edges in the loss function where the edges come from the boundaries of the segmentation results. Talker *et al.* Talker et al. (2022) introduce a depth edges loss to enforce sharp edges in the estimated depth maps. Several other works Ummenhofer et al. (2017); Xian et al. (2020); Wang et al. (2019) attempt to design gradient-based loss functions to improve the details. Xian *et al.* Xian et al. (2020) develop an edge-guided sampling strategy to form a pair-wise ranking loss for enhancing the edge depth.

These methods mainly focus on how to utilize the edge information as an optimization constraint, *i.e.*, use the edges in the loss function. Our approach instead proposes to explicitly use the edge maps as input and fuse the estimated depth from different sources to improve the final depth.

**Depth refinement**    Miangoleh *et al.* Miangoleh et al. (2021) propose a depth refinement method to merge depth from different resolutions and patches via exploiting the effect of the receptive field of the network on high-frequency details prediction. However, this method requires high-quality input images, and its performance would significantly degrade when the input images contain slight noise. Similarly, Dai *et al.* Dai et al. (2023) introduce another merging-based method, which is still sensitive to the image noise as it attempts to merge high-frequency details from high-resolution depth maps into low-resolution depth map images.

Unlike previous methods that rely on high-resolution input images to reserve the details in estimated depth, we focus on the edge information itself for producing high-quality edge depth.

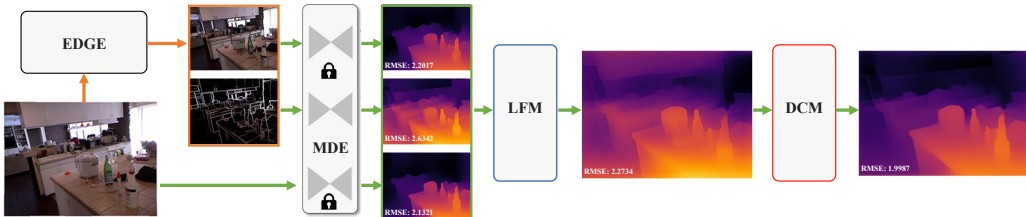

Figure 2: Pipeline of ECFNet. Given an image, ECFNet first extracts the edge map and computes the edge-highlighted image by removing edge pixels. These three images (including the original image) are fed into a frozen MDE network to predict initial depth maps as well. Subsequently, the initial depths are fused using LFM. Finally, the DCM is used to reduce the errors in fused depth from LFM and improve the depth consistency between the final depth and initial depth.

## 3 METHOD

In this section, we demonstrate our proposed method in detail. The overall pipeline of ECFNet is introduced in Sec. 3.1. The proposed hybrid edge detection strategy is illustrated in Sec. 3.2. The core parts of ECFNet (*i.e.*, LFM and DCM) are introduced in Sec. 3.3 and Sec. 3.4 correspondingly.

### 3.1 OVERVIEW

As analyzed in Sec. 1, using the edge information is critical to predicting high-quality depth maps with accurate edges. To this end, we introduce an edge-aware consistency fusion network, named ECFNet, to explicitly fuse the depth of edge maps with the corresponding depth of RGB images. The data flow of ECFNet is shown in Fig. 2. Given a single RGB image $I$, ECFNet first extracts the edge map $I_e$ via using the hybrid edge detection strategy (Sec. 3.2). The corresponding edge-highlighted RGB image $I_{eh}$ is obtained by deleting the edge pixels in $I$. After that, $I$, $I_e$, and $I_{eh}$ are used to infer initial depth $D$, $D_e$ and $D_{eh}$ using the base MDE model, which can be arbitrary pre-trained existing MDE networks. As aforementioned, the $D_e$ has clear edges but lacks reasonable spatial structure due to the absence of texture and shadow in $I_e$. On the contrary, the edges in $D$ and $D_{eh}$ are not clear enough, but their overall structure is more accurate. As a result, the $D$, $D_e$, and $D_{eh}$ are fused by LFM to integrate their advantages. The fused depth $D_{fuse}$ in LFM is not flawless. To fix the errors introduced by wrong spatial structure, inconsistent depth range, and depth distribution between the initial depth (*i.e.*, $D$, $D_e$ and $D_{eh}$) and the $D_{fuse}$, ECFNet utilizes DCM to update the $D_{fuse}$. Consequently, DCM maintains the high-frequency details and recovers the fine structure in the final produced depth.

### 3.2 EDGE DETECTION STRATEGY

As shown in Fig. 3, improving the quality of edge maps could enrich the details in the depth map. However, it is non-trivial to obtain high-quality edge maps. Conventional edge detection algorithms, *e.g.*, Sobel operator Kittler (1983), tend to produce unwanted artifacts and details, such as irregular texture edges, consequently leading to a blurred depth map. Learning-based edge detection methods He et al. (2019); Pu et al. (2022); Su et al. (2021) could generate cleaner edges, but we experimentally find that the edges predicted are always a few pixels away from the accurate position. Hence in this work, we fuse the learning-based BDCN edges He et al. (2019) with Sobel edges Kittler (1983) by calculating their geometric mean for the sake of clean and sharp edges.

Furthermore, inspired by BMD Miangoleh et al. (2021), we increase the input resolution appropriately to enhance the accuracy of the estimated edge map. Concretely, we first divide the image into nine uniform patches, and upsample the patches to high-resolution ones. Then these patches are fed into the BDCN He et al. (2019) network to extract edge maps, which will be fused with the Sobel Kittler (1983) edge maps, and finally downsampled to the original resolution. Such a simple patch strategy could generate more precise BDCN He et al. (2019) edge maps from the high-resolution input with less memory comsumption. In Sec. 4.4, we compare our hybrid edge detection strategy with other methods, and the results indicate our fused edges could fit the accurate edges more closely compared to methods only relying on neural networks. Please see more results in the Appendix A.7.

Figure 3: Comparison of different edge maps and their corresponding depth maps. (a) the RGB image, (b) the Sobel edge map Kittler (1983), (c) our edge map, and (d) the ground truth edge map. As the quality of the edge map improves, the corresponding depth map can capture more depth details (ignoring structural distortion).

## 3.3 LAYERED FUSION MODULE

As mentioned in Sec. 3.1, the initial depth $D$, $D_e$, and $D_{eh}$ possess complementary advantages: $D_e$ contains clear edges and details information but lacks accurate spatial structure, $D$ and $D_{eh}$ have better overall structure but their edges are not clear enough. Such findings inspire us to fuse $D$, $D_e$, and $D_{eh}$ to leverage their respective strengths for producing high-quality depth maps with both clear edges and reasonable structure.

Specifically, we utilize guided filter He et al. (2010) to fuse the different combinations of $D$, $D_e$, and $D_{eh}$ to find an optimal combination. Through a comprehensive comparison (detailed results and analysis in Sec. 4.4), the combination of $(D + D_e + D_{eh})$ is optimal in terms of edge details and overall metrics.

To ensure the fused depth reserves high-frequency details, we propose a layered fusion module, named LFM, to perform the fusion of triad ($D$, $D_e$ and $D_{eh}$). The overall fusion process consists of two stages: $D_e$ and $D_{eh}$ are first fused, then $D$ is fused with the output of the first stage to produce the final result.

As we can use different fusion methods for different stages, we conduct adequate experiments in Sec. 4.4 to determine the final design of LFM. More specifically, we first use guided filter He et al. (2010) to get a fused depth from $D_e$ and $D_{eh}$ as the guided filter reverses the high-frequency details of edge-related depth maps. For the second fusion stage, we design a lightweight convolution network to fuse the depth obtained by guided filters with the original depth $D$. This network uses two CNN-based branches to extract the features of input depths, then uses a shallow CNN-based decoder to fuse the features and output the fused depth.

To train the aforementioned network, we follow the training paradigm proposed in BMD Miangoleh et al. (2021) and GDF Dai et al. (2023). Specifically, we first use DPT Ranftl et al. (2021) to infer the initial depths of the input RGB images with the size of 384×384 and 1024×1024, and then we resize all the initial depths to the size of 1024×1024 to generate training input pairs. The pseudo label of each pair is obtained by using guided filters He et al. (2010) to fuse the pairing depth. Following SGR Xian et al. (2020), we use a gradient domain ranking loss to reserve more details, and use the same depth loss with LeRes Yin et al. (2021b) to improve the scale consistency between the fused depth and the input depths.

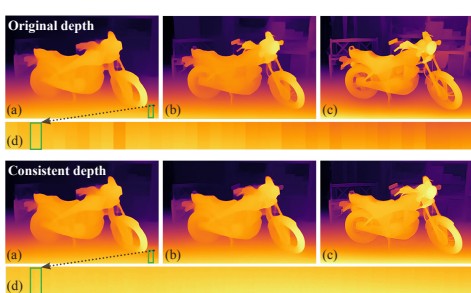

Figure 4: Visualized depth results before and after DCM. (a)-(c) represent the depth maps with different resolutions, while (d) displays depth slices of the green regions in the sample sequence. DCM helps significantly reduce the inconsistency between the input depth maps.

## 3.4 DEPTH CONSISTENCY MODULE

As analyzed in Sec. 1 and Sec. 3.1, fusing multiple depth maps from different inputs, (*i.e.*, original image, edge map, edge-highlighted image) is a stiff task, which is faced with two main challenges: (i) the depth of edge map contains clear edges but suffers from poor spatial structure, (ii) the depth range and distribution of different depth inputs are inconsistent. These challenges result in imperfect fused depth from LFM. To handle this problem, we propose the DCM to fix the errors left by LFM and improve the depth consistency between the final depth and the input depths.

| Method | DIODE Vasiljevic et al. (2019) | | | | IBims-1 Koch et al. (2018) | | | | NYU-v2 Silberman et al. (2012) | | | |
|---|---|---|---|---|---|---|---|---|---|---|---|---|
| | SqRel↓ | ESR↓ | EcSR↓ | ORD↓ | SqRel↓ | ESR↓ | EcSR↓ | ORD↓ | SqRel↓ | ESR↓ | EcSR↓ | ORD↓ |
| (a) CNN-based Ranftl et al. (2022); Yin et al. (2021b) | 7.289 | 7.173 | 7.211 | 0.412 | 0.653 | 0.664 | 0.661 | 0.364 | 0.470 | 0.489 | 0.479 | 0.306 |
| (b) GDF Dai et al. (2023) | 7.304 | 7.372 | 7.402 | 0.423 | 0.625 | 0.645 | 0.651 | 0.377 | 0.432 | 0.440 | 0.438 | 0.295 |
| (c) BMD Miangoleh et al. (2021) | 7.450 | 7.631 | 7.696 | 0.443 | 0.642 | 0.669 | 0.672 | 0.387 | 0.489 | 0.506 | 0.499 | 0.311 |
| (d) Ours (Edge($B$)) | 7.275 | 7.370 | 7.355 | 0.420 | 0.895 | 0.953 | 0.942 | 0.424 | 0.635 | 0.651 | 0.654 | 0.421 |
| (e) Ours (Edge-highlighted ($C$)) | **7.206** | 7.126 | 7.178 | 0.402 | 0.666 | 0.660 | 0.655 | 0.358 | 0.480 | 0.489 | 0.490 | 0.352 |
| (f) Ours ($GF(A+C)$) | 7.321 | 7.419 | 7.401 | 0.419 | 0.656 | 0.646 | 0.653 | 0.366 | 0.454 | 0.483 | 0.478 | 0.299 |
| (g) Ours ($GF(A+B)$) | 7.314 | 7.409 | 7.399 | 0.424 | 0.693 | 0.710 | 0.706 | 0.371 | 0.477 | 0.485 | 0.488 | 0.349 |
| (h) Ours ($GF(B+C)$) | 7.273 | 7.198 | 7.204 | 0.427 | 0.703 | 0.721 | 0.715 | 0.401 | 0.463 | 0.471 | 0.468 | 0.325 |
| (i) Ours ($GF(A+F_{BC})$) | 7.342 | 7.405 | 7.399 | 0.430 | 0.674 | 0.683 | 0.671 | 0.372 | 0.470 | 0.487 | 0.479 | 0.309 |
| (j) Ours-w/o DCM ($LFM(A+F_{BC})$) | 7.354 | 7.371 | 7.345 | 0.432 | 0.679 | 0.683 | 0.677 | 0.379 | 0.466 | 0.457 | 0.459 | 0.334 |
| (k) Ours-BDCN ($LFM(A+F_{BC})$) | 7.313 | 7.222 | 7.209 | 0.409 | 0.541 | 0.540 | 0.548 | 0.349 | 0.402 | 0.396 | 0.394 | 0.275 |
| (l) Ours-Sobel ($LFM(A+F_{BC})$) | 7.276 | 7.016 | 7.071 | 0.399 | **0.533** | **0.521** | 0.517 | **0.333** | 0.418 | 0.398 | 0.392 | 0.290 |
| (m) Ours-Full ($LFM(A+L_{BC})$) | 7.252 | 7.373 | 7.415 | 0.407 | 0.693 | 0.689 | 0.679 | 0.389 | 0.456 | 0.464 | 0.459 | 0.336 |
| (n) Ours-Full ($LFM(A+F_{BC})$) | 7.236 | **6.997** | **7.035** | **0.394** | 0.538 | 0.494 | 0.511 | 0.329 | **0.391** | **0.378** | **0.375** | **0.254** |
| (a) Transformer-based Ranftl et al. (2021) | 7.629 | 7.572 | 7.597 | 0.451 | 0.680 | 0.670 | 0.666 | 0.386 | 0.533 | 0.554 | 0.552 | 0.347 |
| (b) GDF Dai et al. (2023) | 7.522 | 7.542 | 7.551 | 0.449 | 0.613 | 0.620 | 0.624 | 0.375 | 0.531 | 0.532 | 0.535 | 0.335 |
| (c) BMD Miangoleh et al. (2021) | 7.651 | 7.660 | 7.662 | 0.454 | 0.692 | 0.723 | 0.719 | 0.381 | 0.560 | 0.582 | 0.579 | 0.356 |
| (d) Ours (Edge($B$)) | 7.974 | 8.114 | 8.094 | 0.479 | 0.970 | 0.953 | 0.962 | 0.407 | 0.786 | 0.802 | 0.795 | 0.391 |
| (e) Ours (Edge-highlighted ($C$)) | **7.457** | 7.536 | 7.605 | 0.443 | 0.702 | 0.713 | 0.721 | 0.367 | 0.609 | 0.621 | 0.624 | 0.329 |
| (f) Ours ($GF(A+C)$) | 7.522 | 7.622 | 7.615 | 0.448 | 0.691 | 0.710 | 0.714 | 0.366 | 0.501 | 0.506 | 0.507 | 0.294 |
| (g) Ours ($GF(A+B)$) | 7.609 | 7.652 | 7.697 | 0.450 | 0.700 | 0.715 | 0.711 | 0.363 | 0.581 | 0.579 | 0.577 | 0.320 |
| (h) Ours ($GF(B+C)$) | 7.514 | 7.641 | 7.711 | 0.443 | 0.718 | 0.728 | 0.721 | 0.376 | 0.514 | 0.520 | 0.522 | 0.329 |
| (i) Ours ($GF(A+F_{BC})$) | 7.640 | 7.831 | 7.905 | 0.457 | 0.686 | 0.692 | 0.689 | 0.368 | 0.521 | 0.520 | 0.517 | 0.313 |
| (j) Ours-w/o DCM ($LFM(A+F_{BC})$) | 7.556 | 7.560 | 7.588 | 0.446 | 0.692 | 0.690 | 0.688 | 0.369 | 0.598 | 0.594 | 0.592 | 0.304 |
| (k) Ours-BDCN ($LFM(A+F_{BC})$) | 7.579 | 7.342 | 7.412 | 0.440 | 0.521 | 0.516 | 0.518 | 0.340 | 0.490 | 0.488 | 0.485 | 0.288 |
| (l) Ours-Sobel ($LFM(A+F_{BC})$) | 7.523 | 7.260 | 7.315 | 0.438 | **0.518** | 0.499 | 0.502 | 0.335 | 0.501 | 0.496 | 0.495 | 0.287 |
| (m) Ours-Full ($LFM(A+L_{BC})$) | 7.537 | 7.570 | 7.575 | 0.455 | 0.623 | 0.620 | 0.622 | 0.372 | 0.549 | 0.551 | 0.553 | 0.319 |
| (n) Ours-Full ($LFM(A+F_{BC})$) | 7.463 | **7.202** | **7.224** | **0.436** | 0.520 | **0.490** | **0.495** | 0.337 | **0.489** | **0.468** | **0.470** | **0.280** |

Table 1: Quantitative experimental results. **Bold** figures indicate the best and underlined figures indicate the second best. We test our method on three public datasets and select three *SOTA* MDE models Ranftl et al. (2021; 2022); Yin et al. (2021b) to test the effectiveness of our method (we choose two CNN-based MDE method and employ LeRes Yin et al. (2021b) in DIODE Vasiljevic et al. (2019)). We also show the experimental results of the ablation studies in this table. Note that $F_{BC}$ is equal to $GF(B+C)$ which denotes fusing $B$ and $C$ using guided filters He et al. (2010), and $L_{BC}$ is equal to $LFM(B+C)$ which means fusing $B$ and $C$ with LFM. The methods of (d)-(i) do not use the DCM while the methods of (k)-(n) use it. *Ours-BDCN* and *Ours-Sobel* indicate our full method with only BDCN edges He et al. (2019) and Sobel edges Kittler (1983).

Specifically, the DCM takes as inputs the initial fused depth $D_{fuse}$ from LFM, the initial original depth $D$ from original images $I$, and outputs the updated depth $D_{out}$, which can be represented as:

$$D_{out} = D_{fuse} + DCM(D_{fuse}, D). \tag{1}$$

According to Eq. 1, the DCM attempts to learn a depth residual to update the $D_{fuse}$ to minimize the inconsistency between $D_{fuse}$ and $D$. To ensure the computing efficiency, we implement DCM with shallow convolution layers and residual blocks He et al. (2016). The overall architecture of DCM consists of an encoder and a decoder with skip connections between them.

Although the structure of DCM is concise, it is non-trivial to train DCM due to the lack of appropriate training data. To this end, we develop a self-supervised training paradigm to train DCM. Specifically, for each image $I$ in the HRWSI dataset Xian et al. (2020), we generate five images $\{I_s\}_{s=1}^5$ with different resolutions ranging from 384×384 to 1024×1024 as a group of training data. Then we use the pre-trained fixed DPT Ranftl et al. (2021) to produce initial depth maps corresponding to the aforementioned training images. Before computing the loss, the initial depth maps will be resized and cropped into the same resolution as $\{D_s\}_{s=1}^5$. As the high-resolution depth maps contain more high-frequency details, we use the same depth domain loss as MiDaS Ranftl et al. (2022), which constrains the similar depth domain, and a depth consistency loss as:

$$\mathcal{L}_C = \sum_{s=2}^5 \left\| M_s \left( D_s - \hat{D}_{s-1} \right) \right\|_1, D_s \Leftarrow D_s + DCM(D_s, D_{s-1}), \tag{2}$$

where $M_s = \exp(-\alpha|V_s - \hat{V}_{s-1}|^2)$ represents the occlusion weight between $D_s$ and $\hat{D}_{s-1}$, $\alpha$ is set to 50. $\hat{D}_{s-1}$ is obtained by warping the $D_{s-1}$ to $D_s$ according to the *depth color* optical flow $F_{s\Rightarrow s-1}$ between $V_s$ and $V_{s-1}$, where the $V_s$ and $V_{s-1}$ are visualized color images corresponding to $D_s$ and $D_{s-1}$. $\hat{V}_{s-1}$ is obtained by warping $V_{s-1}$ to $V_s$ according to the $F_{s\Rightarrow s-1}$. We use the FlowNet2 Ilg et al. (2017) to compute the backward flow $F_{s\Rightarrow s-1}$ between $V_s$ and $V_{s-1}$.

As shown in Fig. 4, we compare the visualized depth maps before and after DCM. Compared with the initial depth maps, the output depth maps possess significantly superior consistency.

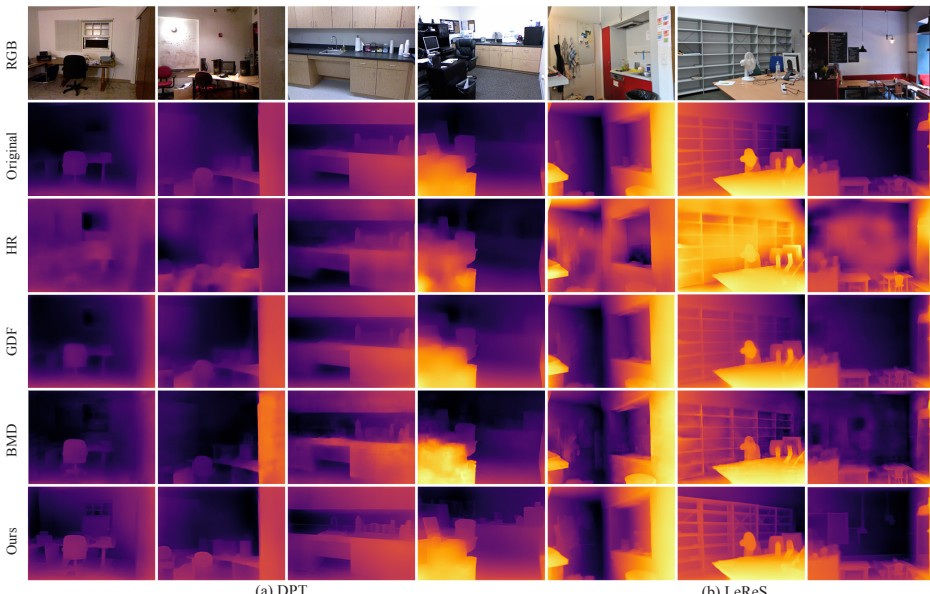

Figure 5: Visualization comparison with base models (DPT Ranftl et al. (2021) and LeReS Yin et al. (2021b)) and related fusion methods (BMD Miangoleh et al. (2021) and GDF Dai et al. (2023)) on IBims-1 Koch et al. (2018) and NYU-v2 Silberman et al. (2012) datasets. *Original* indicates the depth from base models, *HR* indicates the depth of high-resolution inputs from base models. The left four columns use DPT Ranftl et al. (2021) as the base MDE model, and the right three columns use LeRes Yin et al. (2021b).

## 4  EXPERIMENTS

### 4.1  DATASETS AND METRICS

We evaluate our method in three commonly used datasets, DIODE Vasiljevic et al. (2019), IBims-1 Koch et al. (2018), and NYU-v2 Silberman et al. (2012). We adopt the SqRel, AbsRel and RMSE, as the standard evaluation metrics, and ordinal error (ORD) Xian et al. (2020) to reveal finer details when evaluating with degraded images. We also use Edge Square Relative error (ESR and EcSR) to evaluate the edge depth quality. More details about these metrics are provided in the Appendix A.3. Besides, since depth suffers from scale ambiguity, we follow the standard depth evaluation protocol to align the predicted depth and the ground truth depth using the least squares method.

### 4.2  EXPERIMENTS ON PUBLIC DATASETS

To evaluate our proposed method, we test and compare ECFNet with other related methods using the aforementioned datasets. In Tab. 1, we use DPT Ranftl et al. (2021), LeRes Yin et al. (2021b), and MiDaS Ranftl et al. (2022) as base MDE models, and compare ECFNet with base models and another two *SOTA* depth fusion methods, (*i.e.*, BMD Miangoleh et al. (2021) and GDF Dai et al. (2023)). According to the quantitative results in (a)-(c) and (n) of Tab. 1, ECFNet significantly improves the overall quality of initial depth from the base models, especially in NYU-v2 dataset, where our method achieves around $8\%$ and $17\%$ improvement compared to DPT Ranftl et al. (2021) and MiDaS Ranftl et al. (2022) respectively in terms of the SqRel metric. When compared with *SOTA* fusion methods (BMD Miangoleh et al. (2021) and GDF Dai et al. (2023)), ECFNet also obtains a remarkable performance gap.

In Fig. 5, we compare the visualized depth results of base MDE models (DPT Ranftl et al. (2021) and LeRes Yin et al. (2021b)) and related depth fusion methods (BMD Miangoleh et al. (2021) and GDF Dai et al. (2023)) on IBims-1 Koch et al. (2018) and NYU-v2 Silberman et al. (2012) datasets. The predicted depth of ECFNet contains much clearer edges and better overall structure than depth maps from other methods. Following the common practice, we test the base models with high-resolution input images for better details, but the results contain serious artifacts and fewer details than our results.

| Method | IBims1-Noise-gaussian Koch et al. (2018) | | | | IBims1-Blur-gaussian Koch et al. (2018) | | | |
| --- | --- | --- | --- | --- | --- | --- | --- | --- |
| | DPT Ranftl et al. (2021) | | MiDaS Ranftl et al. (2022) | | DPT Ranftl et al. (2021) | | MiDaS Ranftl et al. (2022) | |
| | SqRel↓ | ESR↓ | SqRel↓ | ESR↓ | SqRel↓ | ESR↓ | SqRel↓ | ESR↓ |
| Base Model | 0.8116 | 0.8753 | 0.7729 | 0.8531 | 0.7412 | 0.8124 | 0.7159 | 0.7412 |
| GDF Dai et al. (2023) | 0.7744 | 0.9012 | 0.7160 | 0.7943 | 0.7196 | 0.7743 | 0.7602 | 0.8025 |
| BMD Miangoleh et al. (2021) | 0.8344 | 0.8719 | 0.8032 | 0.8952 | 0.7602 | 0.8432 | 0.7421 | 0.7922 |
| Ours-Sobel | 0.6836 | 0.6692 | **0.6524** | 0.6371 | 0.6837 | 0.6654 | 0.6203 | 0.6019 |
| Ours | **0.6774** | **0.6431** | 0.6607 | **0.6311** | **0.6021** | **0.5947** | **0.6118** | **0.5835** |

Table 2: Quantitative comparisons of the degraded images of IBims1-Noise Koch et al. (2018) dataset.

To further evaluate the trade-off between the inference efficiency and the quality of the depth map produced by our method, we increase the input resolution of base models while maintaining the same runtime of approximately 100ms. As shown in Tab. 3, we present compelling evidence that our method outperforms the baseline model significantly under equal runtime conditions.

Table 3: Quantitative comparison results of inference efficiency, our method outperforms other methods for the same inference time.

| Method | Per-frame Runtime | IBims-1 Ranftl et al. (2021) | | DIODE Ranftl et al. (2022) | |
| --- | --- | --- | --- | --- | --- |
| | | SqRel↓ | ESR↓ | SqRel↓ | ESR↓ |
| DPT Ranftl et al. (2021) | $\simeq 31ms$ | 0.689 | 0.702 | 7.702 | 7.721 |
| LeRes He et al. (2010) | $\simeq 31ms$ | 0.624 | 0.630 | 7.431 | 7.474 |
| DPT Ranftl et al. (2021) | $\simeq 100ms$ | 0.839 | 0.844 | 7.991 | 7.931 |
| LeRes He et al. (2010) | $\simeq 100ms$ | 0.744 | 0.765 | 7.730 | 7.776 |
| GDF Dai et al. (2023) | $\simeq 185ms$ | 0.612 | 0.619 | 7.522 | 7.542 |
| BMD Miangoleh et al. (2021) | $\simeq 14750ms$ | 0.691 | 0.722 | 7.651 | 7.660 |
| Ours-DPT | $\simeq 100ms$ | 0.519 | 0.489 | 7.463 | 7.202 |
| Ours-LeRes | $\simeq 100ms$ | **0.452** | **0.434** | **7.235** | **6.997** |

Table 4: Quantitative comparison of our method with image restoration methods Zamir et al. (2020).

| Method | IBims1-noise-gaussian-001 Koch et al. (2018) | | | | | |
| --- | --- | --- | --- | --- | --- | --- |
| | AbsRel↓ | ESR↓ | RMSE↓ | $\delta_1$ ↑ | ORD↓ | EcSR ↓ |
| DPT Ranftl et al. (2021) | 0.156 | 0.832 | 3.352 | 0.785 | 0.551 | 0.826 |
| Denoising Zamir et al. (2020) | 0.152 | 0.710 | 3.170 | 0.808 | 0.553 | 0.714 |
| Ours | **0.141** | **0.664** | **3.081** | **0.812** | **0.548** | **0.670** |
| Method | IBims1-blur-gaussian-1-7783 Koch et al. (2018) | | | | | |
| | AbsRel↓ | ESR↓ | RMSE↓ | $\delta_1$ ↑ | ORD↓ | EcSR ↓ |
| DPT Ranftl et al. (2021) | 0.148 | 0.751 | 3.228 | 0.803 | 0.613 | 0.747 |
| SR Zamir et al. (2020) | 0.140 | 0.702 | 3.052 | 0.808 | 0.605 | 0.699 |
| Ours | **0.133** | **0.589** | **2.941** | **0.814** | **0.591** | **0.594** |

## 4.3 EXPERIMENTS ON DEGRADED IMAGES

In this subsection we test ECFNet with the degraded images to evaluate its robustness. Concretely we compare ECFNet with other *SOTA* methods on *IBims1-noise-gaussian-001* and *IBims1-blur-gaussian-1-7783* data sequences Koch et al. (2018), where images are corrupted by the Gaussian noise and Gaussian blur, respectively.

Tab. 2 reports the quantitative results. ECFNet still outperforms its counterparts under degraded datasets, and the performance of ECFNet on degraded images is even on par with that of original methods on normal images. Then we visualize the depth maps predicted from degraded images by the base model and ECFNet in Fig. 6. It could be found that the depth maps of ECFNet do not deteriorate significantly with image

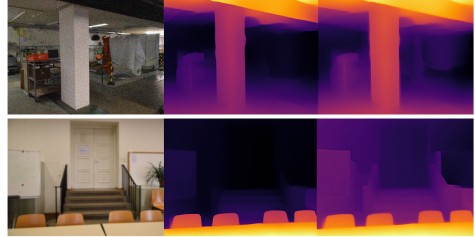

Figure 6: Depth visualization on degraded images of IBims1 Koch et al. (2018) dataset. Each triplet consists of the degraded RGB image, the corresponding depth predicted by DPT Ranftl et al. (2021) and our method.

degradation, but retain no artifacts in noisy images and keep clear in blur images. This is probably because ECFNet relies on edge maps rather than high-resolution input to enhance the high-frequency details, and thus is less likely to be affected by image degradation due to the robust edge strategy.

However, when faced with noisy and blur images, a straightforward solution is to adopt the restoration algorithms as the image preprocessing strategies. Therefore, we further compare ECFNet with the solution above. As shown in Tab. 4, ECFNet surpasses these restoration methods under all six metrics even without specific designs for the degradation. As a result, ECFNet turns out to be a novel method to estimate depth maps with high-frequency details on low-quality and high-noise images.

## 4.4 ABLATION STUDIES

**Hybrid edge detection strategy** As ECFNet takes the edge map as input, the quality of the edge map would affect the final results. In this section, we conduct ablation studies to validate the effectiveness of the hybrid edge detection strategy proposed in this paper.

Concretely, we compare our hybrid edge detection strategy with the conventional edge detection algorithm Sobel operator Kittler (1983) and learning-based edge detection method BDCN He et al. (2019). Firstly we focus on the quality of the edge. As shown in Fig. 7, Sobel edges contain undesired internal texture edges, while BDCN He et al. (2019) edges are coarse in high-frequency regions. On the contrary, our edges are aligned with the actual object edges better. Then we further evaluate the effects of the three kinds of edges. According to (k), (l), and (n) of Tab. 1, the hybrid edge detection strategy shows clear superiority over the other two methods. Hence we could conclude that the final results benefit from better edges, and we adopt the hybrid edge detection strategy in this work.

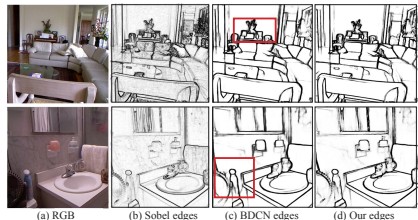

Figure 7: Visualization comparison of different edges. Our edge maps contain fewer texture details and are closer to the real edge.

**Candidate designs of LFM** The target of LFM is to fuse the different depth maps of the original image, the corresponding edge map, and the edge-highlighted image. As introduced in Sec. 3.3, there are several different candidate designs for LFM. In Tab. 1, we test and compare the different schemes of LFM on four public datasets. First, we compare the different combinations with 2 or 3 components in (f)-(i) of Tab. 1. In this group of experiments, we uniformly used the same guided filter (*GF*) for all the methods, and the results indicate that utilizing all of the three depth components helps achieve the best performance. Second, we test the different fusion manners in (i)-(j) and (m)-(n) of Tab. 1. The results of (i) and (j)

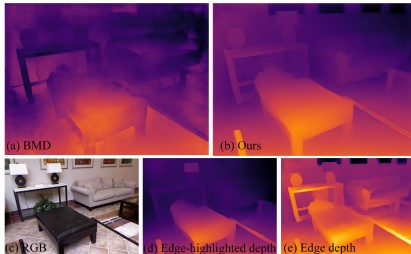

Figure 8: Visualization comparison of different fusion methods.

show that using the network of LFM in the second fusion stage produces better results than using guided filter He et al. (2010), while the results of (m) and (n) indicate using the guided filter in the first fusion stage performs better. Thus we choose the method of (n) as our final scheme. Fig. 8 compares the LFM with the existing fusion method BMD Miangoleh et al. (2021), where the result shows LFM produces fewer artifacts and clear edges, implying the LFM is effective.

**Effectiveness of DCM** DCM aims at minimizing the inconsistency between different depth inputs and producing high-quality depth with clear edges and reasonable structure. In Tab. 1, we test ECFNet with and without DCM using four public datasets. According to the results of (j) and (n) of Tab. 1, DCM plays an important role in improving the quality of final depth. Without DCM, though LFM could generate a clear depth edge, the final overall depth structure is undesirable. As a consequence, DCM is efficacious and indispensable.

## 5 LIMITATIONS

Though ECFNet achieves great performance on various test datasets, there are still several limitations as follows: 1. Compared to the base models or other lightweight MDE models, ECFNet requires more runtime and memory for inference. 2. The performance of ECFNet relies on the quality of the predicted edge maps, but there still remains room for improvement in the edge detection.

## 6 CONCLUSSION

In this paper, we propose ECFNet to estimate high-quality monocular depth with accurate edges and details using a single RGB image. The overall pipeline of ECFNet is designed based on our exploration that the image edge information is critical to reserve high-frequency content. In ECFNet, we first use the hybrid edge detection strategy to get the edge map and edge-highlighted image, which will be fed into a pre-trained MDE network together with the original RGB image to infer the initial depth. Then we propose a layered fusion module (LFM) to fuse the aforementioned three initial depths, and the fused depth will further be optimized by our proposed depth consistency module (DCM) to form the final estimation. As a result, ECFNet achieves significantly superior performance compared with other related methods on four public datasets. We hope that our approach will inspire more robust depth estimation algorithms and facilitate practical downstream applications.

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

# A    APPENDIX

## A.1    EDGE INFORMATION

In the main paper, we believe that edge information plays a crucial role in depth estimation. With the development of large-scale models Saharia et al. (2022); Ramesh et al. (2022); Rombach et al. (2022) and edge-controlled image generation techniques Zhang & Agrawala (2023), it has become possible to generate images of the same scene with different styles but the same edge structure, which provides an opportunity to further validate the significance of edge information. To this end, we employ edge maps as structural control conditions and generate a large number of synthetic images using ControlNet Zhang & Agrawala (2023) and stable diffusion models Rombach et al. (2022). These images have almost identical edge information but distinct textures and materials. Subsequently, we utilize the state-of-the-art depth estimation model DINOv2 Oquab et al. (2023) to generate corresponding depth maps for each image, and remarkably, these depth maps exhibit highly consistent structural information, as demonstrated in Fig. 9. To further verify that this phenomenon is attributed to the identical edge structure information, as shown in Fig. 10, we select 1/4 parts of 4 images (blue, yellow, green, red) from each group of images and combine them. Then, we predict the depth of the combined images. The generated depth maps still preserve high structural similarity.

To further quantify the difference among depth maps generated from these synthetic images, we leverage ControlNet Zhang & Agrawala (2023) and stable diffusion models Rombach et al. (2022) to generate 50 distinct synthetic images based on edge maps, and randomly arrange them to obtain 50 sets of depth maps. We compute the absolute error between each pair of depth maps, as illustrated in Fig. 11, where we randomly sample several groups of results (approximately 200 groups of data per group) to demonstrate their absolute error quantification indicators. The errors between these depth maps are negligible due to their highly consistent edge structure information, which further corroborates the crucial role of edge information in depth estimation.

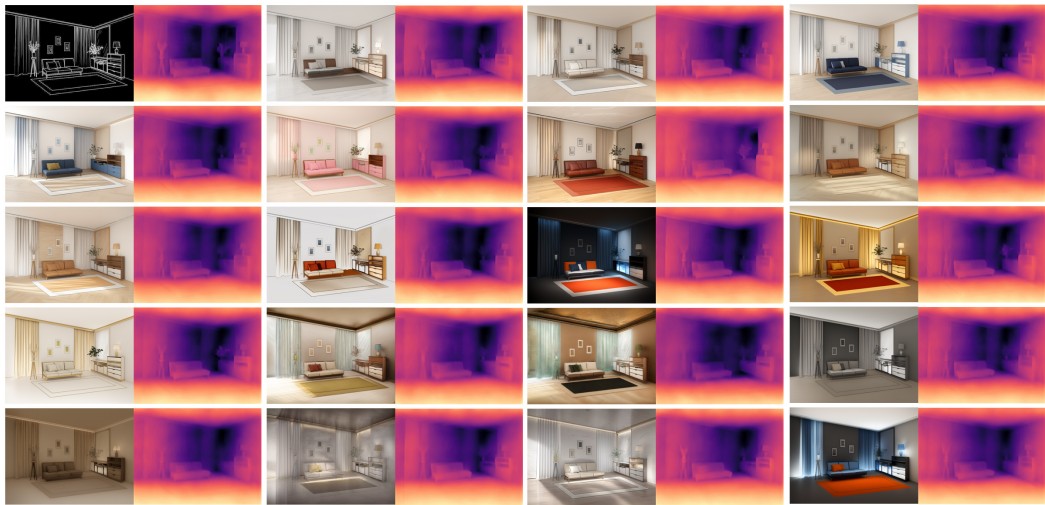

Figure 9: We present visualization results of depth maps predicted from various input images. Each image pair consists of an RGB image and the depth map predicted by DINOv2 Oquab et al. (2023). The structural information of these output depth maps is almost identical, as the corresponding input RGB images have very similar edge information.

## A.2    POTENTIAL APPLICATIONS

Due to the diverse categories of artistic creations, existing depth estimation techniques struggle to accurately obtain depth maps for artistic images, which poses challenges for downstream tasks such as 3D photos Shih et al. (2020) and bokeh. However, our analysis shows that edges play a crucial role in depth estimation. By combining ControlNet Zhang & Agrawala (2023) and diffusion models Rombach et al. (2022) to generate cross-domain synthetic images, we can preserve the edge

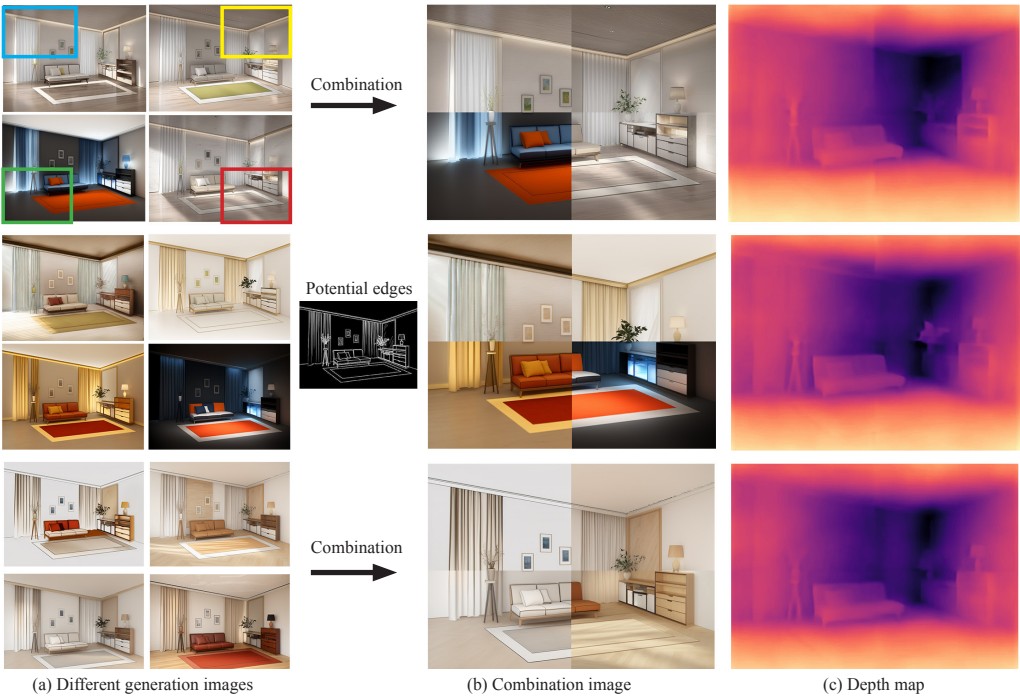

(a) Different generation images      (b) Combination image      (c) Depth map

Figure 10: We form a group of four RGB images and extract 1/4 from each generation image to combine into a new image. The depth maps predicted from these combination images are almost identical because they share similar edge structure information.

structure of the original artistic images and achieve cross-domain style transfer. As shown in Fig 12, the resulting depth maps refined by our ECFNet have more details and clear edges by inputting the new synthetic image, leading to better performance in downstream tasks and promoting practical applications. We hope our findings and methods can drive the development of cross-domain image applications and inspire more edge-based approaches.

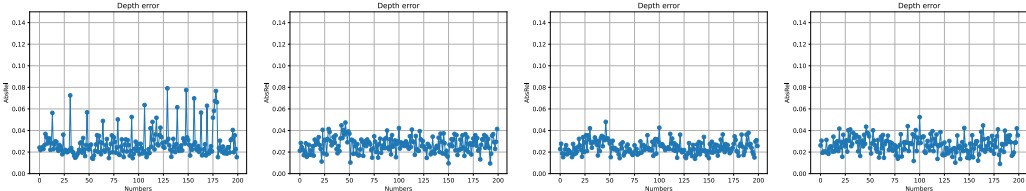

Figure 11: We show the absolute errors between each random pair of depth maps. The errors between these depth maps are negligible. Specifically, we show the unstable error results in the first figure to indicate that the diffusion model Rombach et al. (2022) will generate some bad synthetic images in rare cases, thus making the corresponding depths also differ from the others.

## A.3 EVALUATION METRICS

We conduct our experiments in the depth space, and therefore the results predicted by MiDaS Ranftl et al. (2022) and DPT Ranftl et al. (2021) are processed by the formula $D = D_{max} - D$. We utilize the commonly applied depth estimation metrics, which are defined as follows:

- Absolute relative error (AbsRel): $\frac{1}{n} \sum_{i=1}^{n} \frac{|d_i - d_i^*|_1}{d_i^*}$;

- Square relative error (SqRel): $\frac{1}{n} \sum_{i=1}^{n} \frac{(d_i - d_i^*)^2}{d_i^*}$;

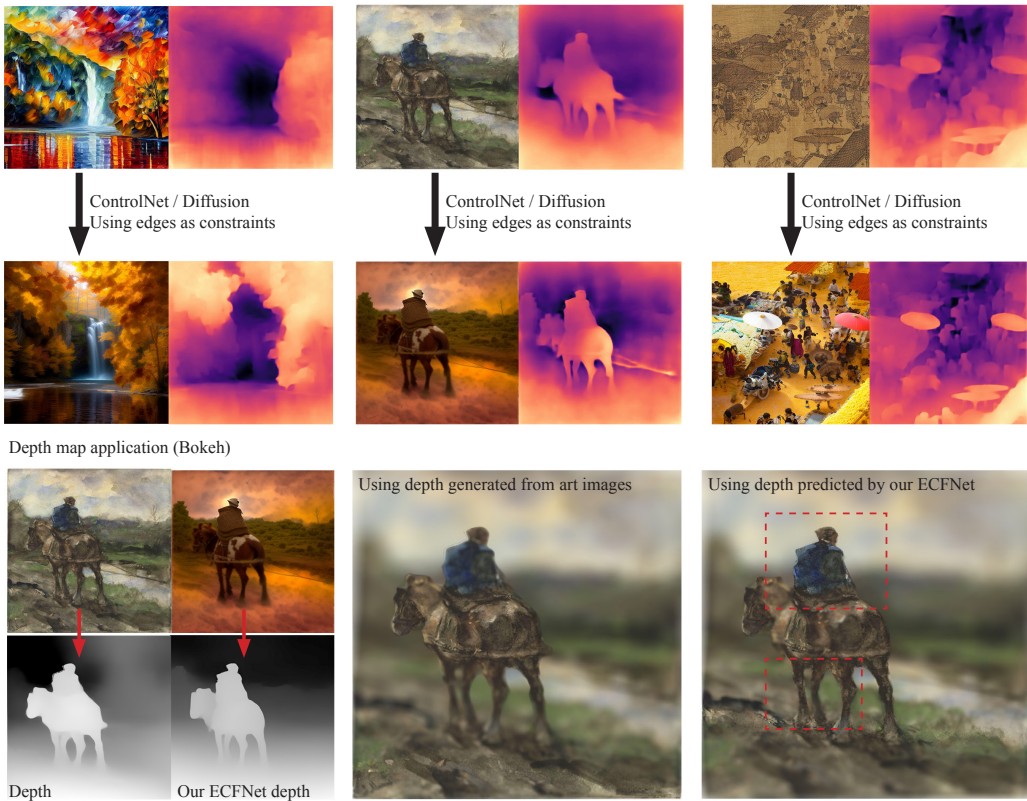

Figure 12: We can preserve the edge structure of the original artistic images by generating cross-domain synthetic images. Subsequently, the synthetic image with edge information is fed into our ECFNet to predict accurate depth, enabling a number of downstream applications.

- Edge Square relative error (ESR / EcSR): $ESR = \frac{1}{N}\sum_{i=1}^{N} E_i * \frac{(d_i - d_i^*)^2}{d_i^*}, EcSR = \frac{1}{N}\sum_{i=1}^{N} Ec_i * \frac{(d_i - d_i^*)^2}{d_i^*}$ whert $E_i$ is the our edge binary mask pixel, $Ec_i$ means the canny edge Canny (1986) binary mask pixel (the high and low thresholds are set to 100 and 200, respectively). $N$ is the number of pixels.

- Root mean squared error (RMSE): $\sqrt{\frac{1}{n}\sum_{i=1}^{n}(d_i - d_i^*)^2}$;

- Accuracy with threshold $t$: Percentage of $d_i$ such that $max(\frac{d_i}{d_i^*}, \frac{d_i^*}{d_i}) = \delta_1 < 1.25$;

- Ordinal error (ORD) Xian et al. (2020): $\frac{\sum_i \omega_i \mathbb{I}(\ell_i \neq \ell_{i,\tau}^*(p))}{\sum_i \omega_i}$, where $\omega_i$ is a weight to set to 1, ORD use the function with sample strategy between $\ell_i$ and $\ell_{i,\tau}^*(p)$. ORD is a general metric for evaluating the ordinal accuracy of a depth map.

where $n$ denotes the total number of pixels, $d_i$ and $d_i^*$ are estimated and ground truth depth of pixel $i$, respectively.

**Loss Functions in Layer Fusion Module (LFM)**    For the loss function in the depth domain, we use the same loss as LeRes Yin et al. (2021b):

$$L_{\text{ILNR}} = \frac{1}{N}\sum_{i}^{N} \left| d_i - \overline{d}_i^* \right| + \left| tanh(d_i/100) - tanh(\overline{d}_i^*/100) \right|,$$

where $\overline{d}_i^* = (d_i^* - \mu_{\text{trim}})/\sigma_{\text{trim}}$ and $\mu_{\text{trim}}$ and $\sigma_{\text{trim}}$ are the mean and the standard deviation, we remove the maximum and minimum $10\%$ of depth values in the depth map. $d$ is the predicted depth, and $d^*$

Figure 13: The network structure of our layer fusion module (LFM).

Figure 14: The network structure of our depth consistency module (DCM). $R$ represents the residual blocks.

is the ground truth depth map. Given a set of sampled point pairs $\mathcal{P} = \{[p_{i,0}, p_{i,1}], i = 1, \ldots, N\}$, we use the same structure-guided ranking loss as SGR Xian et al. (2020) in the gradient domain:

$$\mathcal{L}_{\text{rank}}(\mathcal{P}) = \frac{1}{N} \sum_i \phi(p_{i,0} - p_{i,1}),$$

More details about the sample strategy can be found in their paper Xian et al. (2020).

## A.4 IMPLEMENTATION DETAILS

**Layer Fusion Module (LFM)** We utilize an encoder-decoder network with 10 layers to increase the training and inference resolution to $1024 \times 1024$. The network is similar to u-net Ronneberger et al. (2015), to better capture the edge gradient information, we divide the input into two streams: one for the depth with more complete scene structure, and the other stream for the depth with more edge datails. We use a kernel of size $3 \times 3$. Group normalization Wu & He (2018) and leaky ReLUs Paszke et al. (2017) are applied except the last layer. We show the network structure in Fig. 13. During training, we implement our model using PyTorch and train it with the AdamW solver Loshchilov & Hutter (2017) for 9,000 iterations with a learning rate of 1e-4. To obtain our training data, we generate 3,000 training data pairs from the HRWSI Xian et al. (2020) dataset using DPT Ranftl et al. (2021), where the low resolution is $384 \times 384$ and the high resolution is $1024 \times 1024$. Pseudo labeled data with high-frequency and less depth artifacts can be generated by the method of fusing them through the guided filter. The window radius of the guided filter is set to one-twelfth of the depth map width, and the edge threshold is 1e-12.

**Depth Consistency Module (DCM)** The DCM utilizes an encoder-decoder architecture that is similar to Lai *et al.* Lai et al. (2018). Specifically, the encoder consists of two downsampling strided convolutional layers, which are then followed by five residual blocks. The decoder consists of two transposed convolutional layers, and we incorporate skip connections from the encoder to the decoder. We show the network structure in Fig. 14. Instance Normalization Ulyanov et al. (2017) is applied except the last layer. Finally, we apply a Tanh layer to ensure that the output is within the range from -1 to 1 after the decoder. We implement our model using PyTorch and train it with the Adam solver Diederik & Jimmy (2015) for 50,000 iterations with a learning rate of 1e-4. During training, we use a batch size of 4 and randomly crop training data to 192×192.

## A.5 EDGE STRATEGY

BMD Miangoleh et al. (2021) presents a comprehensive analysis of the network's receptive field size, focusing on its application to depth estimation. We can increase the resolution of the input image to obtain more precise depth estimation results. Interestingly, we also find that the resolution increase can benefit the generation of edge maps. Specifically, the higher resolution enables better localization of object boundaries, resulting in sharper and more detailed edges. This observation suggests that resolution enhancement can be a valuable approach for improving learning-based edge detection methods.

As a result, we appropriately increase the input resolution to enhance the accuracy of the estimated edge map,. Specifically, we first divide the image into nine uniform patches and upsample each patch to three times its original resolution. Then, these patches are fed into the BDCN network He et al. (2019) to extract edge maps, which are subsequently fused with Sobel edge maps Kittler (1983).

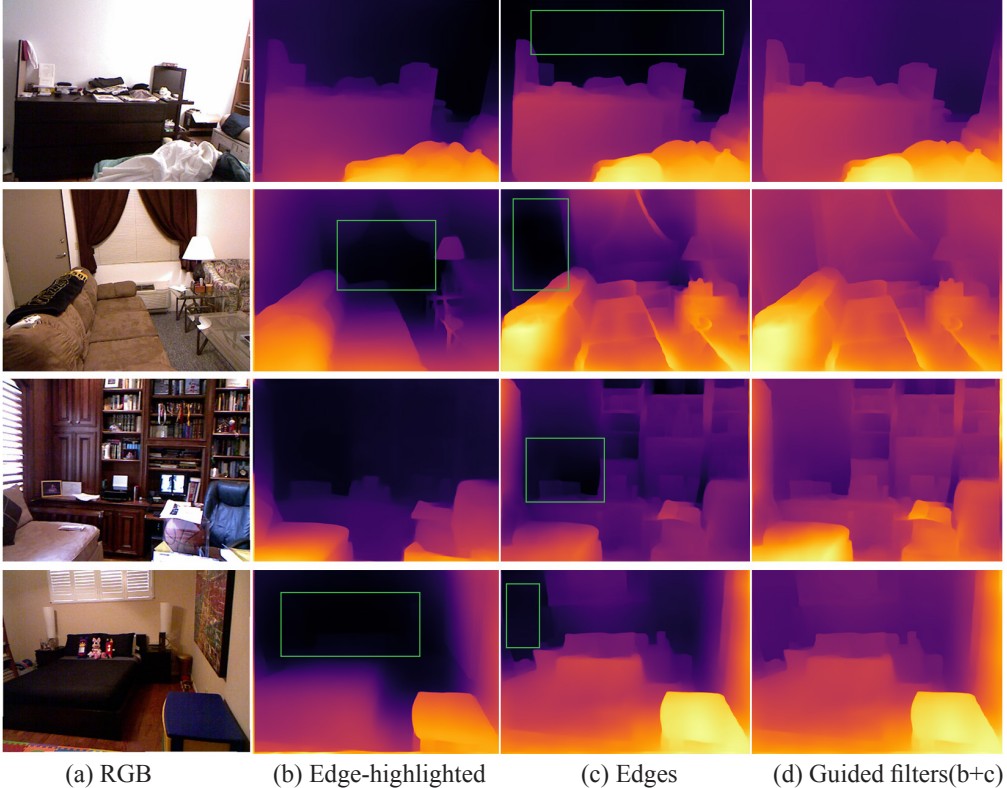

|     (a) RGB     |  (b) Edge-highlighted  |  (c) Edges  |  (d) Guided filters(b+c)  |

Figure 15: We show the fusion results of the guided filter, and the green boxes indicate the artifact areas in the depth map estimated from the edge map and the edge-highlighted image. The guided filter can effectively eliminate these artifacts.

The fusion process is represented as $\sqrt[N]{E_b \times E_s}$, where $N$ is set to 2 to enhance contrast on both sides of the edge pixels, and $E_b$ and $E_s$ represent the BDCN edge and Sobel edge, respectively. This approach effectively eliminates hard edges and improves the overall smoothness and naturalness of the resulting image. Finally, we downsample the generated edge map to the original resolution. As shown in Fig. 16, we provide more edge maps generated by our edge strategy.

### A.6    WHY USE GUIDED FILTERS?

The goal for the LFM is to seamlessly merge the complementary information coming from the depth maps which separately predicted by the original images, the edge maps and edge-highlighted images. As mentioned in the main paper, these depths possess complementary advantages. Actually, the depth map predicted by the edge map has the details we most expect to fuse, but it loses the overall structure of the scene and generates artifacts which cannot solve by directly using weighted averanging or Poisson blending. Guided filtering He et al. (2010)can be an effective technique for our task, especially when the input depth maps have different scene structures and details that need to be preserved in the fused image. We show the experimental comparison and visualization results of these two methods in Tab. 2 and Fig. 15.

### A.7    MORE VISUALIZATION RESULTS

We show more visualization results comparing the state-of-the-arts depth fusion methods with our method. As shown in Fig. 17, we achieve higher depth accuracy and obtains more edge details.

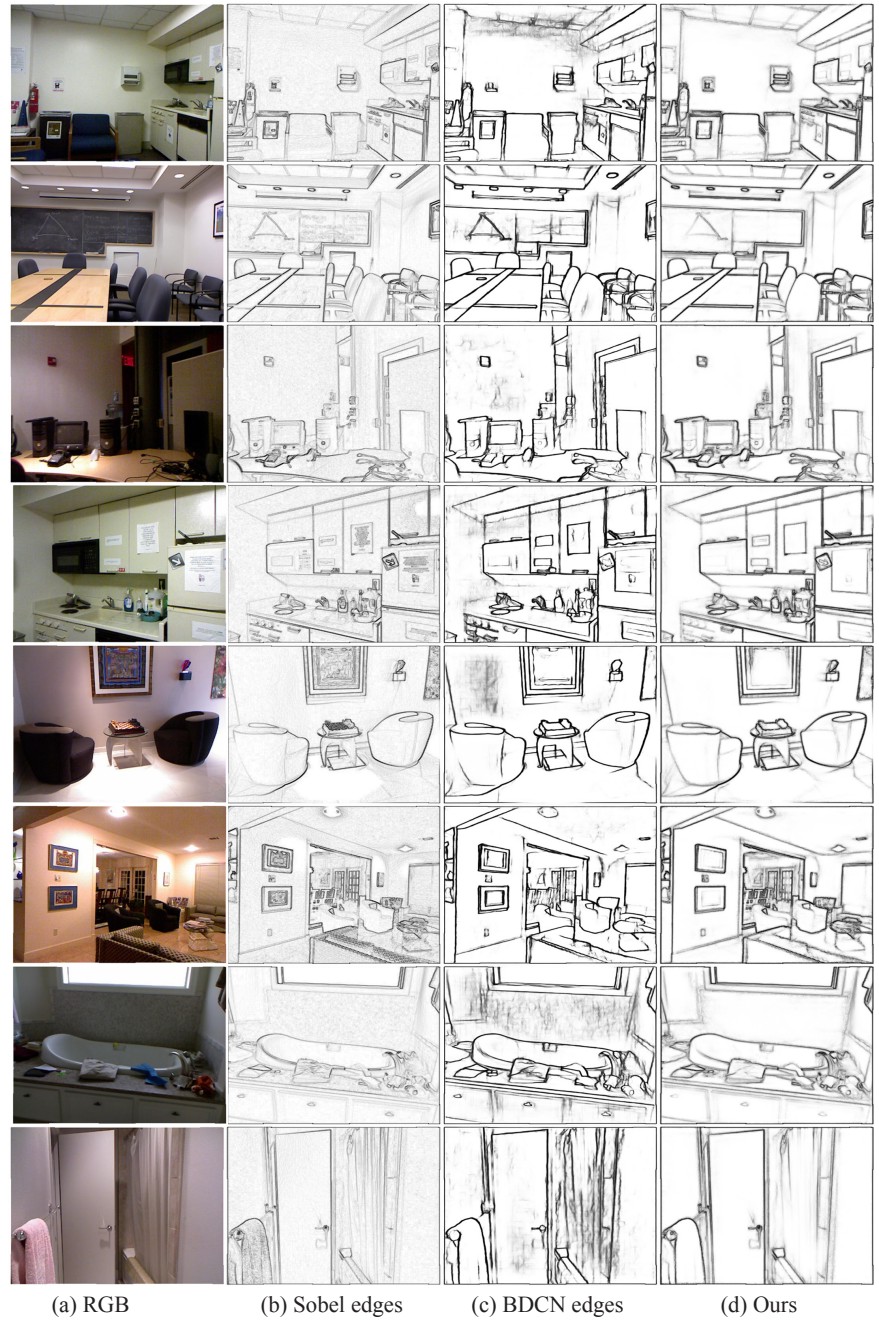

(a) RGB       (b) Sobel edges       (c) BDCN edges       (d) Ours

Figure 16: We show more visualization results of the edge map. The edge maps generated by our edge strategy are cleaner and closer to the actual edge maps.

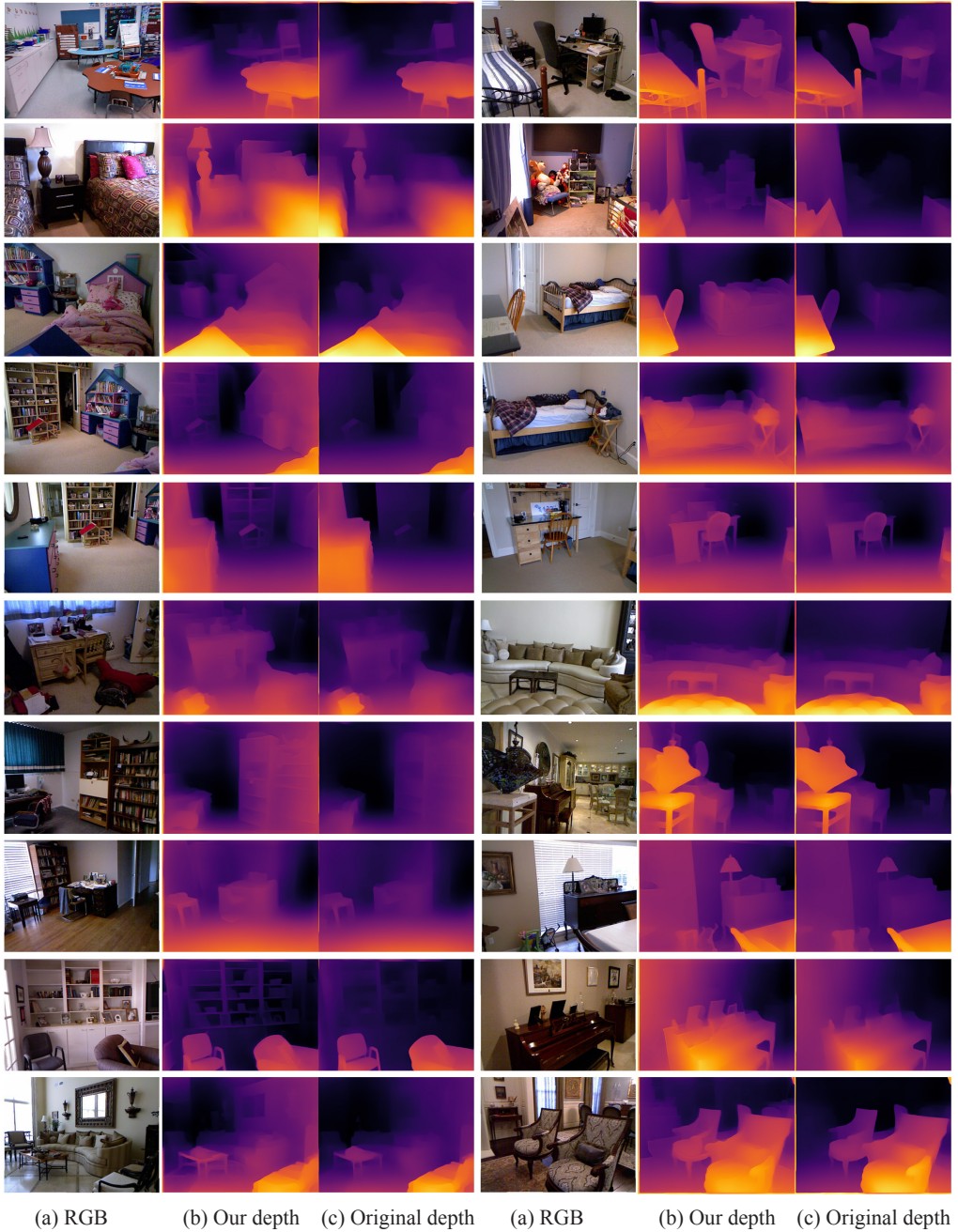

(a) RGB     (b) Our depth   (c) Original depth     (a) RGB     (b) Our depth   (c) Original depth

Figure 17: We provide more depth visualization. Each triplet consists of the RGB image, the depth predicted by our method, and the depth predicted by DPT Ranftl et al. (2021).

