# OpenReview forum: "The Devil is in the Edges: Monocular Depth Estimation with Edge-aware Consistency Fusion"
_ICLR.cc/2024/Conference — ICLR 2024 Conference Withdrawn Submission_

### Official Review · Reviewer_ZRn8 · 2023-10-20

**Soundness:** 2 fair
**Presentation:** 2 fair
**Contribution:** 2 fair
**Rating:** 5
**Confidence:** 4

**Summary:**

This paper addresses monocular depth estimation with a special focus on image edges. Based on observations of the network under different input versions (original input image, edges, and input image without edges) the authors build an ensemble model to combine multiple depth estimates in a multi-stage manner. In addition to utilizing a pre-trained DPT model, three additional networks are introduced into the depth estimation pipeline: one for edge detection, one for depth fusion, and the last one for depth refinement.

**Strengths:**

1. Combining classical and deep learning methods for edge detection is an interesting approach that can be useful for several tasks.

2. Good quantitative results.

3. Extensive ablation studies.

**Weaknesses:**

1. Unclear method.
     a) Figure 2 does not depict the inner workings of the network. For instance, the residual connection in Eq. 1 is not shown in Figure 1.
     b) Un-defined variables. V after equation 2 is not clearly defined. How are visualized color images obtained?
     c) Optical flow is introduced from nowhere. What is the optical flow used for if this is single view depth estimation?
     d) Is the optical flow network needed during inference as well?

2. Improvements for depth estimates are unclear due to the low resolution of the provided results. Have you considered showing your performance improvements with surface normals or 3D point clouds?

3. Robustness against noisy images is an interesting side effect, but it might come from the edge detection network, not from the authors' contributions per se.

4. Considerable additional running time is addressed in the limitations, but the added numbers of parameters are not commented on.

5. Potentially low novelty due to the use of pre-trained models that are simply run several times with added edge, fusion, and refinement blocks.

**Questions:**

1. Several times in the paper you use the word "reserve", but it seems you meant "preserve." is that correct?

---

### Official Review · Reviewer_bKTL · 2023-10-28

**Soundness:** 3 good
**Presentation:** 3 good
**Contribution:** 2 fair
**Rating:** 3
**Confidence:** 3

**Summary:**

This paper studies the edge information in monocular depth estimation (MDE). It begins with an analysis of depth estimation from images, edge maps, and edge-highlighted maps. Then, it designs three strategies, including hybrid edge detection strategy, layered fusion module, and depth consistency module. Experiments on three datasets show the effectiveness.

**Strengths:**

- The motivation is analyzed clearly. It is easy for readers to understand the method.

- This paper is well-written and easy to follow.

- The performance is good.

- The ablation is detailed.

**Weaknesses:**

- The core idea of this paper is the fusion of depth maps from multiple inputs, including original RGB, edge map, and edge-highlighted map, which I find very limited novelty.

- The proposed three strategies: hybrid edge detection strategy, layered fusion module, and depth consistency module also contain no new thing. Regarding the hybrid edge detection strategy, it fuses two edge maps one from a learning-based method and one from the Sobel algorithm. Regarding the layered fusion module, it exploits the previous guided filter. Regarding depth consistency module, it is a trivial variant of previous methods.

- Experiments on outdoor datasets, such as KITTI, is required.

Overall, it is a good technical report, but needs great improvements for a top conference.

**Questions:**

see weaknesses.

---

### Official Review · Reviewer_sHHy · 2023-11-05

**Soundness:** 3 good
**Presentation:** 3 good
**Contribution:** 3 good
**Rating:** 5
**Confidence:** 5

**Summary:**

This paper presents a new method for monocular depth estimation. It identifies a limitation in existing approaches, which often fail to provide accurate depth maps along edges. Building upon new observations, the paper introduces a novel network and learning strategy for predicting depth maps. The proposed work consists of three innovative modules: a hybrid edge detection module (designed to enhance edge quality), a layered fusion module (for integrating depth maps from multiple sources), and a depth consistency module (used to refine the fused depth map and maintain consistency across different resolutions). The study conducts extensive experiments on various modules and evaluates their performance on three different datasets to establish their efficacy.

**Strengths:**

+ The overall contribution of this paper is a novel fusion-based approach for monocular depth estimation. It utilized fusion in various stages of its architectural pipeline in general.

+ The first contribution involves the practical implications of employing various input modalities in existing MDE methods. This paper meticulously sought to comprehend how image edges influence learning-based monocular depth map estimation. It demonstrated that current state-of-the-art MDE models perform sub-optimally in scenarios characterized by low contrast, long distances, and small edges. This was substantiated through both visual and quantitative assessments by predicting depth maps from three different input modalities: traditional RGB, edge map, and edge-highlighted RGB (achieved by removing edge pixels from the RGB). The findings led to the conclusion that when a depth map is inferred from an MDE model using only an edge map, it yields a depth map with clear edges but fails to capture the overall scene structure. However, the overall scene structure can be recovered without fine-edge details when the MDE model is employed to predict a depth map from RGB and edge-highlighted RGB inputs. Based on these two observations, this paper proposed a fusion-based method that combines depth maps produced from three distinct sources.

+ The second contribution pertains to the technical approach used to incorporate their observations in fusing complementary depth maps predicted from three modalities: traditional RGB, edge map, and edge-highlighted RGB. This approach involves three steps. 1) The first technical contribution encompasses an improvement in the edge detection strategy using a hybrid approach that combines edges estimated from a classical edge predictor (Sobel operator) and a data-centric approach (BDCN). This hybrid approach, in itself, represents another fusion method. 2) The second technical contribution is the fusion method that combines three depth maps generated by a frozen state-of-the-art MDE model, such as DPT or MiDAS. This fusion method includes strategies for selecting the number of maps, their order of combination, and the fusion method itself, which can involve techniques like guided filtering (a prior work) and a U-Net style convolutional neural network (CNN). 3) The third and final technical contribution consists of an additional module designed to refine the fused depth map and minimize inconsistencies between different depth inputs, particularly at varying resolutions. Experimental evidence demonstrates that this final refinement is indispensable for enhancing the overall quality of the depth map.

+ The paper conducts comprehensive experiments on multiple datasets to demonstrate the effectiveness of the proposed fusion-based MDE approach. Experiments include testing in the presence of noise, degraded images, the efficiency of the proposed method, an ablation study, and a comparison against other competing methods.

**Weaknesses:**

- While the introduction of a new hybrid approach for edge discovery is promising, qualitatively, it demonstrates the superiority of the hybrid edge detection approach over either of its two constituent approaches from which edges are detected. However, it lacks a quantitative evaluation of the edge maps.

- The proposed approach utilizes a fusion of multiple initial depth maps from MDE approaches, such as DPT or MiDAS. These methods are also demonstrated as baselines for comparisons; therefore, the proposed method is expected to excel in performance, as demonstrated in Table I. Furthermore, the loss functions used in the layered fusion module are also adopted from the two methods (Yin et al. and Xian et al.) against which comparisons are made in Table I. As a result, the proposed method is designed to yield better performance. It would have been better to demonstrate another state-of-the-art method against which the proposed method was neither adopted nor shared in any module.

**Questions:**

Please, see the weakness section.